# Cardiogenic Shock Does Not Portend Poor Long-Term Survival in Patients Undergoing Primary Percutaneous Coronary Intervention

**DOI:** 10.3390/jpm12081193

**Published:** 2022-07-22

**Authors:** Eva Steinacher, Felix Hofer, Niema Kazem, Andreas Hammer, Lorenz Koller, Irene Lang, Christian Hengstenberg, Alexander Niessner, Patrick Sulzgruber

**Affiliations:** Department of Internal Medicine II, Division of Cardiology, Medical University of Vienna, Waehringer Guertel 18-20, 1090 Vienna, Austria; eva.steinacher@meduniwien.ac.at (E.S.); felix.hofer@meduniwien.ac.at (F.H.); niema.kazem@meduniwien.ac.at (N.K.); andreas.hammer@meduniwien.ac.at (A.H.); lorenz.koller@meduniwien.ac.at (L.K.); irene.lang@meduniwien.ac.at (I.L.); christian.hengstenberg@meduniwien.ac.at (C.H.); patrick.sulzgruber@meduniwien.ac.at (P.S.)

**Keywords:** acute coronary syndrome, cardiogenic shock, long-term prognosis, long-term survival, percutaneous coronary intervention, risk stratification

## Abstract

Although a strong association of cardiogenic shock (CS) with in-hospital mortality in patients with acute coronary syndrome (ACS) is well established, less attention has been paid to its prognostic influence on long-term outcome. We evaluated the impact of CS in 1173 patients undergoing primary percutaneous coronary interventions between 1997 and 2009. Patients were followed up until the primary study endpoint (cardiovascular mortality) was reached. Within the entire study population, 112 (10.4%) patients presented with CS at admission. After initial survival, CS had no impact on mortality (non-CS: 23.5% vs. CS: 24.0%; *p* = 0.923), with an adjusted hazard ratio of 1.18 (95% CI: 0.77–1.81; *p* = 0.457). CS patients ≥ 55 years (*p* = 0.021) with moderately or severely impaired left ventricular function (LVF; *p* = 0.039) and chronic kidney disease (CKD; *p* = 0.013) had increased risk of cardiovascular mortality during follow-up. The present investigation extends currently available evidence that cardiovascular survival in CS is comparable with non-CS patients after the acute event. CS patients over 55 years presenting with impaired LVF and CKD at the time of ACS are at increased risk for long-term mortality and could benefit from personalized secondary prevention.

## 1. Introduction

The acute phase of acute coronary syndrome (ACS) is frequently accompanied by severe complications such as reduced systolic ventricular pumping function or arrhythmia, especially ventricular tachycardia (VT) or ventricular fibrillation (VF), which can culminate in cardiogenic shock (CS) [1]. Signs of CS have been described to be present in 4–12% of all ACS cases, with a considerably higher occurrence in ST-segment elevation myocardial infarction (STEMI), reflecting a frequent burden in daily acute cardiac care [2,3,4,5].

CS is defined as the most severe form of acute heart failure resulting in global tissue hypoperfusion and severe peripheral organ-damage [6]. Poor prognosis for the initial survival of patients with CS during ACS is well known, with an in-hospital mortality rate of approximately 40–50% and almost half of the lethal events occurring within the first 24 h of hospitalization [7,8]. Subsequently, patients presenting with CS require early revascularization and immediate intensive care measures to reduce both mortality and morbidity [9,10].

Although the association of CS with poor short-term survival is well established, the overall knowledge about the impact of CS on patient outcomes from a long-term perspective is limited. Most importantly, data on both patient- and therapy-related characteristics that influence the prognosis of CS patients remain unknown but seem of utmost importance in terms of risk stratification and secondary prevention in the era of personalized medicine.

Considering the gap of evidence, we aimed to investigate the impact of CS on long-term survival in patients after ACS. Moreover, we intended to assess prognostic values for cardiovascular mortality during long-term follow-up in CS patients who survived the initial ACS event.

## 2. Materials and Methods

### 2.1. Study Population

A detailed study protocol has already been described elsewhere [11]. In short, a total of 1173 patients presenting with ACS who were subjected to coronary angiography in the high-volume cardiac catheterization laboratory of the University Hospital Vienna, Vienna General Hospital (Austria), between January 1997 and December 2009, were included in this study. Guidelines of the European Society of Cardiology (ESC) were used to precisely define acute myocardial infarction (AMI), including the distinction between STEMI and non-ST-segment elevation myocardial infarction (NSTEMI) [12,13]. Patient charts were screened for signs of CS at hospital admission, which were defined as signs of end-organ hypoxia due to impaired cardiac output. Hemodynamic instability in terms of hypotension with systolic blood pressure (BP) < 90 mmHg or the need of vasopressors, development of fatal arrhythmia or cardiac arrest, and clinical presentation with blood-flow centralization, oliguria, altered mental status, and lactate > 2.0 mmol/L were rated as CS signs. No specific exclusion criteria were applied for patient enrolment to achieve an unselected patient population. The study protocol was accepted in compliance with ethical principles of the Declaration of Helsinki by the ethics committee of the Medical University of Vienna (EK 159/2011).

### 2.2. Data Acquisition and Follow-Up

At the time of acute hospitalization, physicians and nurses from the emergency department determined the patient’s hemodynamic status by evaluating clinical signs, electrocardiography (ECG) analysis, and BP measurements. In addition, a routine echocardiography was performed by specialists directly in the emergency department or after the primary intervention on the cardiology ward. Patients were continuously monitored during a period of three days after the acute event via electronic telemetry. Immediately before and after coronary angiography, blood samples were collected and analysed for routine parameters (i.e., blood cell count with differential, chemistry panels as well as cardiac biomarkers) by local standards of the Department of Laboratory Medicine at the Medical University of Vienna (Austria).

Specially trained chart reviewers collected and evaluated precisely defined patient characteristics. Patient-related data were gathered via the local electronic patient database including intensive care unit (ICU) records of the Medical University of Vienna (Austria).

Standardized follow-up was carried out until January 2017 via the local electronic patient registry. Cardiovascular mortality was defined as the primary study endpoint. Dates and causes of death of all patients were validated by the Austrian death registry (Statistics Austria, Vienna, Austria) to avoid endpoint bias. 

### 2.3. Statistical Analysis

Non-metric variables were described by counts and percentage values and subsequently statistically compared using chi-square tests. Continuous variables were shown as median and interquartile range (IQR) and compared using Mann–Whitney U tests. In addition, Cox-proportional hazard regression analysis was performed to investigate the influence of CS and possible prognostic factors on cardiovascular mortality. Statistical values are presented as hazard ratios (HRs) together with their 95% confidence intervals (Cis). For multivariate analysis, the regression model was adapted to the potential confounding factors age and sex. To graphically illustrate the influence of CS on cardiovascular mortality, Kaplan–Maier charts were plotted. Group-differences were determined by log-rank tests.

For statistical analysis, SPSS Statistics 27 program (IBM SPSS Statistics, NY, USA) was used. All tests were performed with a two-sided statistical significance level of *p* < 0.05.

## 3. Results

The study cohort of 1173 patients with AMI presented with a median age of 57.2 years (IQR 42.0–80.0), a median body mass index (BMI) of 26.5 kg/m^2^ (IQR 24.2–29.6), and a proportion of women of 38.5% (*n* = 451). The fraction of STEMI in the study group was 48.4% (*n* = 563). A total of 122 patients (10.4%) were diagnosed with CS at hospital admission, 88 (72.1%) of whom presented with VT or VF, and 93 (76.9%) required resuscitation before the primary angiography. 

Table 1 shows baseline characteristics of the study population including in-hospital management and primary percutaneous coronary intervention (pPCI)-related data, stratified into hemodynamically stable patients (*n* = 1051; 89.6%) and patients with CS (*n* = 122; 10.4%) during the acute event. A statistical comparison of the study groups indicated comparable age and sex-distribution, proportion of STEMI, and atrial fibrillation (AF). As expected, patients with CS showed low systolic BP (non-CS: 127 mmHg vs. CS: 114 mmHg; *p* < 0.001) on ACS admission as well as impaired left ventricular function (LVF; normal: non-CS: 49.3% vs. CS: 39.8%; mildly reduced: non-CS: 20.2% vs. CS: 10.2%; moderately reduced: non-CS: 20.3% vs. CS: 26.1%; highly reduced: non-CS: 10.2% vs. CS: 23.9%; *p* < 0.001)

Cardiovascular risk factors and comorbidities were significantly less common in patients presenting with CS in contrast to patients free of CS: hypertension (non-CS: 70.1% vs. CS: 52.3%; *p* < 0.001), family history of coronary artery disease (CAD; non-CS: 36.9% vs. CS: 26.0%; *p* = 0.030), history of CAD (non-CS: 22.8% vs. CS: 13.5%; *p* = 0.045), and hypercholesterolemia (non-CS: 68.7% vs. CS: 45.0%; *p* < 0.001). There were more patients with pre-existing chronic heart failure in the CS group (non-CS: 5.0% vs. CS: 10.3%; *p* = 0.018).

Both groups were treated with pPCI (non-CS: 83.8% vs. CS: 84.4%; *p* = 0.854), whereas thrombolytic therapy was more often given to patients with CS (non-CS: 12.8% vs. CS: 24.8%; *p* < 0.001). The severity of CAD and the culprit lesion location showed no differences between the subgroups; however, more left main (LM) stenoses were present in CS patients (non-CS: 3.0% vs. CS: 6.9%). 

In hemodynamically unstable patients, significantly increased cardiac troponin-T (TnT; non-CS: 2.0 µg/L vs. CS: 3.6 µg/L; *p* < 0.001) and creatinine kinase (CK; non-CS: 698.0 U/L vs. CS: 1530.0 U/L; *p* < 0.001) values were evident. Moreover, as expected in hemodynamically compromised patients, serum creatinine values on hospital admission were higher in CS patients than in non-CS patients (non-CS: 1.03 mg/dL vs. CS: 1.14 mg/dL; *p* < 0.001). 

### 3.1. Survival Analysis

For the entire study population, the median duration of hospitalization was nine days (IQR 6–14), with a longer hospital stay in the CS strata (non-CS: 9 days vs. CS: 13 days; *p* < 0.001). As expected, in-hospital mortality in patients presenting with CS during the initial event was significantly higher compared to non-CS patients (non-CS: 3.7% vs. CS: 21.3%; *p* < 0.001). 

Patients were followed until the primary study endpoint was reached. The median follow-up time was 9.0 years (IQR 6.1–11.7), corresponding to 10,545 patient years. During the overall study period, a total of 431 (36.8%) deaths were recorded, of which 318 (73.8%) were cardiovascular. Patients with CS who survived the index hospitalization achieved a long-term mortality comparable to patients without signs of CS (non-CS: 23.5% vs. CS: 24.0%; *p* = 0.923, see Table 2). 

Furthermore, we observed no association of CS on long-term cardiovascular mortality after hospital discharge with a crude HR of 1.02 (95% CI: 0.67–1.57; *p* = 0.928). These results remained stable, even after comprehensive adjustment to potential confounders within the multivariate model with an adjusted HR of 1.18 (95% CI: 0.77–1.81; *p* = 0.457), as shown in Table 3. Results were plotted in Kaplan–Meier survival curves and confirmed via log-rank test (*p* = 0.928; see Figure 1). 

### 3.2. Personalized Prognostication for Long-Term Outcome in Cardiogenic Shock

After presenting with CS during the acute phase of ACS, 96 patients (78.7%), 35 (36.5%) of whom were female, survived until hospital discharge. During long-term follow-up, a total of 23 (24.0%) participants of this primarily surviving group died of cardiovascular causes. 

Table 4 shows a comparison of long-term CS survivors (*n* = 73) and deceased patients (*n* = 23) after survival of the initial hospitalization. Both groups showed similar patient characteristics in terms of female sex (CS survivors: 35.6% vs. CS deceased: 39.1%; *p* = 0.760) and clinical presentation during the acute event with comparable numbers of VT or VF (CS survivors: 74.0% vs. CS deceased: 60.9%; *p* = 0.228) and necessity of cardio-pulmonary resuscitation (CPR) before angiography (CS survivors: 79.2% vs. CS deceased: 65.2%; *p* = 0.174). Moreover, measures of in-hospital management tended to be balanced between groups with similar fraction of fibrinolysis (CS survivors: 26.1% vs. CS deceased: 13.6%, *p* = 0.227) and PCI rates (CS survivors: 86.3% vs. CS deceased: 91.3%, *p* = 0.527).

CS patients over 55 years had an increased risk of cardiovascular death after hospital discharge (CS deceased < 55 years: 39.1% vs. ≥55 years: 60.9%; *p* = 0.017). In addition, we were able to show that patients who experienced a fatal event during long-term follow-up were more likely to present with moderately or severely impaired LVF (CS survivors: 40.0% vs. CS deceased: 73.3%, *p* = 0.021), new-onset AF (CS survivors: 6.8% vs. CS deceased: 21.7%; *p* = 0.042), and chronic kidney disease (CKD; CS survivors: 1.4% vs. CS deceased: 22.7%, *p* < 0.001). 

We observed that age ≥ 55 (adj. HR: 2.82; 95% CI: 1.17–6.78; *p* = 0.021), presence of moderately or severely impaired LVF (adj. HR: 3.46; 95% CI: 1.06–11.21; *p* = 0.039), and CKD (adj. HR: 4.43; 95% CI: 1.38–14.26; *p* = 0.013) were strong and independent predictors for mortality in CS individuals who survived the index event.

## 4. Discussion

The present investigation represents, to the best of our knowledge, the longest follow-up period of patients showing signs of CS during ACS. Although it is well known that ACS patients presenting with CS are at an increased risk for cardiovascular death during the acute phase of the index event, the presented data highlights that if the index event was survived, CS patients reached long-term survival rates comparable to their non-CS counterparts. 

Considering an individualized secondary prevention after ACS complicated by CS, patients over 55 years that present with impaired LVF and CKD seem to be at increased risk for fatal events from a long-term perspective. Therefore, these patients could potentially benefit from intensified follow-up measures in order to improve secondary prevention.

### 4.1. Cardiogenic Shock in Acute Coronary Syndrome

In the present investigation, we were able to demonstrate that patients developing CS during ACS tended to be younger and healthier with fewer pre-existing comorbidities such as arterial hypertension, hypercholesteremia or type-II diabetes mellitus. In contrast, patients with previously known CAD appeared to have a survival benefit. This paradox could be explained by the fact that ACS often presents the primary manifestation of CAD. Patients with new-onset or short-term CAD have not (yet) developed sufficient collateral vessels as they do in chronic hypoxic myocardium [14]. Acute coronary occlusion in ACS with insufficient collateral blood flow results in extensive myocardial tissue damage and, accordingly, poor contractility can be expected in the acute event. Both factors favour the occurrence of arrhythmias due to altered refractory periods and scar areas, in particular fatal ventricular arrhythmias [15]. Hence, the presence of collateral arteries due to pre-existing CAD appears to have a protective effect against the development of CS in the early phase of AMI, as they can potentially minimize the infarct size in ACS, which is also the most plausible assumption for the observations in our study population.

As might be expected, we detected a trend toward higher rates of LM stenosis in the CS collective compared to hemodynamically stable patients. The LM provides approximately 84% of the blood flow supplying the left ventricle, therefore, acute stenosis in this vessel results in a correspondingly large and jeopardized infarct area. Especially in the left ventricle, decreased perfusion of large myocardial areas possibly manifest as contractility restrictions, decrease in cardiac output and, consequently, end-organ damage with shock symptoms [16]. The kidney, as a highly perfusion sensitive organ, responds early to even minor changes in the cardiovascular system. Reduced peripheral blood flow can be well reflected by increased creatinine levels, as observed in our CS population at the time of hospital admission. Creatinine is a well-known sensitive marker of end-organ damage due to low cardiac output or hypovolemia [17]. 

The hypothesis considering a larger infarct area in CS patients is supported by elevated maximum values of cardiac biomarkers CK and TnT in the CS subgroup compared to the non-CS cohort. CK and TnT correlate precisely with the magnitude of the myocardial damage, which has been repeatedly demonstrated and confirmed in multiple imaging studies [18,19,20,21,22]. In addition, significantly elevated initial C-reactive protein (CRP) values in the CS cohort indicate a marked systemic inflammatory response in these patients. Elevated CRP levels represent a prognostically unfavourable situation, as it has been proven to be predictable for increased rates of adverse cardiac events, poor in-hospital outcomes, acute kidney injury, and mortality [23,24,25,26].

### 4.2. Survival of the Initial Acute Event

In the overall study population of 1173 patients with ACS, CS was observed in 122 patients, corresponding to a rate of 10.4%—which is in line with international observations [8,27]. Relatively elevated CS rates in our study cohort can be attributed to the single-centre setting of our study, including only patients of the high-volume cardiac catheterization laboratory of the Vienna University Hospital with extensive treatment options, including mechanical assist devices as Impella and, in the case of our patient enrolment between 1997 and 2009, intra-aortic balloon pump (IABP) and extracorporeal membrane oxygenation (ECMO). 

In consistency with pre-existing evidence, patients presenting with CS during ACS showed significantly increased in-hospital mortality rates compared to non-CS patients. We were able to demonstrate an adjusted HR of 7.45 for in-hospital mortality in CS patients as well as a mortality rate of 21.3% compared to 3.7% for non-CS patients. Similar studies, however, reported mortality rates nearly twice as high at about 35–50% [7,8,28]. Higher survival rates in our patient population may be due to the omission of low-frequency hospitals with fewer treatment possibilities, and less experience and expertise. 

### 4.3. Long-Term Mortality and Identification of Cardiogenic Shock Patients at Risk for Fatal Cardiovascular Events

Age, left ventricular impairment, intubation, systolic BP, and the laboratory parameters lactate and base excess are characteristics mentioned in literature that proved to be associated with poor survival of the initial event [29,30]. However, it is unclear whether these factors also apply to the long-term survival of patients with AMI complicated by CS during the initial event. We were able to extend currently available evidence that patients with CS who survived the acute phase of ACS had similar rates of fatal cardiovascular events to those observed in patients free of CS from a long-term perspective (non-CS: 23.5% vs. CS: 24.0%). 

Considering an individualized secondary prevention after ACS complicated by CS, patients older than 55 years that present with CKD and moderately or severely impaired LVF seem to be at increased risk for fatal events after hospital discharge. We also detected a trend for increased event rates in patients with new-onset AF. The association between increased rates of mortality and higher age, as well as impaired left ventricular ejection fraction (LVEF) has been described in the literature [31,32].

Paradoxically, non-smokers who presented with CS in the acute event seem to have an increased long-term mortality risk compared to smokers. This has already been observed in other studies and can probably be attributed, again, as in patients with CAD, to the recruitment of myocardial collateral vessels. Smoking could promote the formation of collateral vessels due to the induction of chronic hypoxia; however, this has not been fully clarified [33,34].

Especially in the era of personalized medicine, precise patient characterization and the detection of risk factors is of utmost importance to take a step towards individualized treatment strategies. Within this regard, recently published studies demonstrated that machine learning is able to extend traditional statistical analysis in order to identify novel risk factors for ACS-related adverse events [35,36]. The evidence of these recently published trials highlight the feasibility of this method in improving risk prediction and could act as a risk assessment tool in the future. Concerning CS patients at risk for long-term mortality, personalized secondary prevention measures in the form of intensified follow-ups, optimal medical therapy (OMT) or the benefit of implantable cardioverter-defibrillator (ICD) or cardiac resynchronization therapy and defibrillator (CRT-D) implantation need to be evaluated.

### 4.4. Limitations

The major limitation of the present study is the potential selection bias due to the single-centre setting, considering that only patients treated in the local high-frequency cardiac catheterization laboratory of the University Hospital in Vienna were included, thus neglecting low-frequency hospitals. However, the large study cohort and long follow-up time as well as unselected patient sampling reduces the probability of introducing relevant bias.

## 5. Conclusions

With the present investigation, we demonstrated that CS during ACS constitutes a strong prognostic factor for short-term survival—however, the results of this study clearly highlight that if survival of the index event was achieved, CS patients reached comparable long-term survival to their non-CS counterparts. Most importantly, age over 55 years, CKD, and moderately or severely reduced LVF were found to be highly predictive values to identify CS patients at risk for cardiovascular mortality during long-term follow-up. Patients who present with these characteristics may benefit from intensified and individualized secondary prevention measures in the era of personalized medicine.

## Figures and Tables

**Figure 1 jpm-12-01193-f001:**
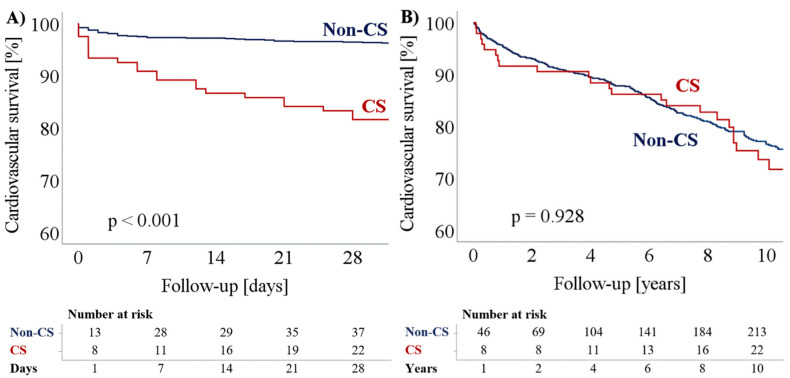
Accumulated cardiovascular survival comparing hemodynamically stable patients to patients with cardiogenic shock, (**A**) during the acute event, *p* < 0.001 and (**B**) after survival of the acute event, *p* = 0.928. CS = cardiogenic shock.

**Table 1 jpm-12-01193-t001:** Baseline characteristics.

	No Cardiogenic Shock (*n* = 1051)	Cardiogenic Shock (*n* = 122)	*p*-Value
**Clinical presentation**			
Age [years], (IQR)	57.7 (42.0–80.0)	53.7 (41.6–73.4)	0.195
Female sex, *n* (%)	410 (39.0%)	41 (33.6%)	0.242
BMI [kg/m^2^], (IQR)	26.6 (24.2–29.6)	26.1 (23.9–29.7)	0.276
STEMI, n (%)	504 (48.4%)	59 (48.4%)	0.991
AF, *n* (%)	92 (8.8%)	16 (13.1%)	0.116
**Hemodynamic status at admission**			
Heart rate [bpm], (IQR)	75 (65–86)	81 (70–99)	**<0.001**
Systolic BP [mmHg], (IQR)	127 (114–141)	114 (100–130)	**<0.001**
Diastolic BP [mmHg], (IQR)	75 (65–83)	68 (60–80)	**<0.001**
LVF			**<0.001**
Normal, *n* (%)	396 (49.3%)	35 (39.8%)	
Mildly reduced, *n* (%)	162 (20.2%)	9 (10.2%)	
Moderately reduced, *n* (%)	163 (20.3%)	23 (26.1%)	
Highly reduced, *n* (%)	82 (10.2%)	21 (23.9%)	
**Risk factors and comorbidities**			
Current smoker, *n* (%)	561 (57.7%)	60 (58.3%)	0.907
Family history of CVD, *n* (%)	348 (36.9%)	26 (26.0%)	**0.030**
Hypertension, *n* (%)	729 (70.1%)	57 (52.3%)	**<0.001**
Diabetes mellitus, *n* (%)	205 (19.8%)	14 (12.8%)	0.078
Hypercholesterolemia, *n* (%)	710 (68.7%)	50 (45.0%)	**<0.001**
Previous AMI, *n* (%)	207 (20.0%)	18 (15.5%)	0.245
Previous CVD, *n* (%)	163 (22.8%)	12 (13.5%)	**0.045**
Chronic heart failure, *n* (%)	52 (5.0%)	12 (10.3%)	**0.018**
CKD, *n* (%)	74 (7.2%)	11 (9.6%)	0.352
**In-hospital management**			
Fibrinolysis, *n* (%)	131 (12.8%)	28 (24.8%)	**<0.001**
PCI, *n* (%)	878 (83.8%)	103 (84.4%)	0.854
No intervention, *n* (%)	141 (13.4%)	16 (13.1%)	0.941
**PCI-related data**			
Vessel disease			0.359
0-CVD, *n* (%)	51 (4.9%)	7 (5.7%)	
1-CVD, *n* (%)	519 (49.4%)	65 (53.3%)	
2-CVD, *n* (%)	285 (27.1%)	24 (19.7%)	
3-CVD, *n* (%)	196 (18.6%)	26 (21.3%)	
Culprit lesion			0.241
LM, *n* (%)	24 (3.0%)	7 (6.9%)	
LAD, *n* (%)	LAD, *n* (%)	397 (49.1%)	53 (52.0%)
CX, *n* (%)	98 (12.1%)	12 (11.8%)	
RCA, *n* (%)	261 (32.3%)	29 (28.4%)	
Diag., *n* (%)	13 (1.6%)	0 (0%)	
Marg., *n* (%)	16 (2.0%)	1 (1.0%)	
Number of stents, (IQR)	1 (1–2)	1 (1–2)	0.205
**Laboratory analysis**			
TnT, max. [µg/L], (IQR)	2.0 (0.6–4.7)	3.6 (1.2–7.0)	**<0.001**
CK, max. [U/L], (IQR)	698.0 (248.5–1695.0)	1530.0 (489.0–4488.0)	**<0.001**
CK-MB, max [U/L], (IQR)	105.5 (51.7–235.4)	245.0 (103.0–567.0)	**<0.001**
LDH, max. [U/L], (IQR)	405.0 (269.0–655.0)	696.0 (452.5–1314.0)	**<0.001**
BNP [pg/mL], (IQR)	1025.0 (296.7–3372.0)	387.8 (182.4–2338.0)	0.098
CRP [mg/dL], (IQR)	0.6 (0.4–1.5)	0.7 (0.5–3.7)	**0.030**
Creatinine, pre-PCI [mg/dL], (IQR)	1.03 (0.89–1.22)	1.14 (0.99–1.53)	**<0.001**
Creatinine, post-PCI [mg/dL], (IQR)	0.99 (0.85–1.19)	1.01 (0.80–1.43)	0.749
HbA1c [%], (IQR)	5.7 (5.4–6.2)	5.6 (5.2–6.3)	0.141

Metric data are displayed as median with the respective interquartile range (IQR) in parenthesis. Discrete data are demonstrated as counts with percentages in parenthesis. For statistical analysis, Mann–Whitney U test was used for metric data and Chi-square test was used for categorial data. Statistical significance is shown in bold values. AF = atrial fibrillation, AMI = acute myocardial infarction, BMI = body mass index, BNP = B-type natriuretic peptide, BP = blood pressure, CK = creatinine kinase, CKD = chronic kidney disease, CK-MB = creatinine kinase—muscle and brain type, CRP = C-reactive protein, CVD = coronary vessel disease, CX = left circumflex artery, Diag. = diagonal branch, HbA1c = haemoglobin A1c, LAD = left anterior descending artery, LDH = lactate dehydrogenase, LM = left main, LVF = left ventricular function, Marg. = marginal branch, PCI = percutaneous coronary intervention, RCA = right coronary artery, STEMI = ST-elevation myocardial infarction, TnT = troponin T.

**Table 2 jpm-12-01193-t002:** Short and long-term outcome.

	No Cardiogenic Shock (*n* = 1051)	Cardiogenic Shock (*n* = 122)	*p*-Value
Duration of hospitalization, (IQR)	9 (6–13)	13 (8–23)	**<0.001**
In-hospital mortality, *n* (%)	39 (3.7%)	26 (21.3%)	**<0.001**
CV-death after hospital discharge, *n* (%)	238 (23.5%)	23 (24.0%)	0.923

Metric data are displayed as median with the respective interquartile range (IQR) in parenthesis. Categorial data are demonstrated as counts with percentages in parenthesis. For statistical analysis, Mann–Whitney U test was used for metric data and Chi-square test was used for categorial data. Statistical significance is shown in bold values. CV = cardiovascular.

**Table 3 jpm-12-01193-t003:** Crude and adjusted effects of cardiogenic shock on cardiovascular in-hospital and long-term mortality.

	Crude HR (95% CI)	*p*-Value	Adjusted HR (95% CI)	*p*-Value
**Cardiovascular in-hospital mortality**				
Cardiogenic shock	6.20 (3.77–10.19)	**<0.001**	7.45 (4.51–12.33)	**<0.001**
**Cardiovascular mortality after hospital discharge**				
Cardiogenic shock	1.02 (0.67–1.57)	0.928	1.18 (0.77–1.81)	0.457

Cox proportional hazard model. The multivariate model was adjusted for age and sex. Statistical significance is shown in bold values. CI = confidence interval, HR = hazard ratio.

**Table 4 jpm-12-01193-t004:** Baseline characteristics with crude and adjusted hazard ratios on long-term cardiovascular mortality for cardiogenic shock patients.

	CS Survivors (*n* = 73)	CS Deceased (*n* = 23)	*p*-Value	Crude HR (95% CI)	*p*-Value	Adjusted HR (95% CI)	*p*-Value
**Clinical presentation**							
Age [years], (IQR)	44.0 (40.5–64.0)	65.1 (41.0–81.0)	**0.030**	1.67 (1.10–2.55)	**0.016**	1.70 (1.11–2.61)	**0.014**
Age ≥ 55 years, *n* (%)	24 (32.9%)	14 (60.9%)	**0.017**	2.62 (1.13–6.08)	**0.024**	2.82 (1.17–6.78)	**0.021**
Female sex, *n* (%)	26 (35.6%)	9 (39.1%)	0.760	0.94 (0.41–2.16)	0.878	1.18 (0.50–2.79)	0.697
BMI [kg/m^2^], (IQR)	25.8 (23.9–29.2)	25.1 (22.5–30.5)	0.742	0.88 (0.56–1.41)	0.608	0.85 (0.52–1.38)	0.539
STEMI, *n* (%)	31 (42.5%)	11 (47.8%)	0.651	1.28 (0.56–2.89)	0.561	1.08 (0.44–2.63)	0.863
AF, *n* (%)	5 (6.8%)	5 (21.7%)	**0.042**	3.34 (1.23–9.06)	**0.018**	2.36 (0.82–6.78)	0.112
**Hemodynamic status at admission**							
Heart rate [bpm], (IQR)	81.0 (70.0–98.5)	77.5 (68.3–99.0)	0.662	0.96 (0.57–1.60)	0.865	1.11 (0.66–1.87)	0.687
Systolic BP [mmHg], (IQR)	119.0 (110.0–130.0)	109.0 (99.0–133.0)	0.120	0.78 (0.52–1.18)	0.239	0.78 (0.54–1.14)	0.197
Diastolic BP [mmHg], (IQR)	70.0 (61.5–80.0)	62.0 (56.0–80.0)	0.269	0.76 (0.50–1.14)	0.187	0.77 (0.52–1.14)	0.192
VT/VF, *n* (%)	54 (74.0%)	14 (60.9%)	0.228	0.60 (0.26–1.38)	0.226	1.00 (0.39–2.59)	0.994
CPR before angiography, *n* (%)	57 (79.2%)	15 (65.2%)	0.174	0.60 (0.25–1.42)	0.244	0.93 (0.37–2.34)	0.874
Moderately and severely red. LVF, *n* (%)	24 (40.0%)	11 (73.3%)	**0.021**	3.71 (1.18–11.65)	**0.025**	3.46 (1.06–11.21)	**0.039**
**Comorbidities**							
Current smoker, *n* (%)	50 (73.5%)	6 (31.6%)	**<0.001**	0.23 (0.09–0.60)	**0.003**	0.25 (0.08–0.81)	**0.021**
Family history of CVD, *n* (%)	22 (33.3%)	3 (15.8%)	0.139	0.36 (0.11–1.24)	0.106	0.34 (0.10–1.18)	0.090
Hypertension, *n* (%)	37 (55.2%)	9 (42.9%)	0.322	0.61 (0.26–1.45)	0.261	0.46 (0.19–1.13)	0.089
Diabetes mellitus, *n* (%)	6 (9.0%)	3 (14.3%)	0.482	1.90 (0.56–6.50)	0.304	1.40 (0.40–4.85)	0.598
Hypercholesterolemia, *n* (%)	36 (52.9%)	8 (36.4%)	0.176	0.50 (0.21–1.19)	0.117	0.45 (0.18–1.12)	0.086
Previous AMI, *n* (%)	7 (10.1%)	4 (18.2%)	0.314	1.43 (0.48–4.23)	0.519	0.67 (0.20–2.24)	0.513
Previous CVD, *n* (%)	5 (9.8%)	2 (11.8%)	0.818	1.01 (0.23–4.41)	0.994	0.70 (0.23–2.16)	0.532
Chronic heart failure, *n* (%)	3 (4.3%)	3 (13.6%)	0.126	2.17 (0.64–7.32)	0.214	1.50 (0.43–5.27)	0.526
CKD, *n* (%)	1 (1.4%)	5 (22.7%)	**<0.001**	6.24 (2.24–17.41)	**<0.001**	4.43 (1.38–14.26)	**0.013**
**In-hospital management**							
Fibrinolysis, *n* (%)	18 (26.1%)	3 (13.6%)	0.227	0.42 (0.12–1.43)	0.165	0.59 (0.17–2.11)	0.418
PCI, *n* (%)	63 (86.3%)	21 (91.3%)	0.527	1.81 (0.42–7.73)	0.424	2.40 (0.55–10.51)	0.246
No intervention, *n* (%)	8 (11.0%)	2 (8.7%)	0.757	0.71 (0.17–3.04)	0.645	0.46 (0.10–2.05)	0.308
**PCI-related data**							
Vessel disease			0.421	0.59 (0.31–1.13)	0.110	0.65 (0.35–1.23)	0.183
0-CVD, *n* (%)	2 (2.7%)	1 (4.3%)					
1-CVD, *n* (%)	37 (50.7%)	15 (65.2%)					
2-CVD, *n* (%)	22 (30.1%)	6 (26.1%)					
3-CVD, *n* (%)	12 (16.4%)	1 (4.3%)					
Culprit lesion			0.925	0.86 (0.55–1.36)	0.520	0.89 (0.57–1.39)	0.609
LM, *n* (%)	4 (6.2%)	2 (10.5%)					
LAD, *n* (%)	33 (50.8%)	9 (47.4%)					
CX, *n* (%)	8 (12.3%)	3 (15.8%)					
RCA, *n* (%)	19 (29.2%)	5 (26.3%)					
Marg., *n* (%)	1 (1.5%)	0 (0.0%)					
**Laboratory Analysis**							
TnT, max. [µg/L], (IQR)	3.0 (1.2–6.5)	3.5 (0.7–6.4)	0.675	0.83 (0.52–1.34)	0.454	0.73 (0.43–1.26)	0.263
CK, max. [U/L], (IQR)	1425.0 (534.8–4515.3)	1266.0 (269.0–3700.0)	0.319	0.82 (0.54–1.23)	0.330	0.90 (0.57–1.43)	0.655
CK-MB, max [U/L], (IQR)	214.0 (81.0–508.0)	407.0 (147.0–653.5)	0.182	1.42 (0.87–2.33)	0.163	1.71 (1.00–2.91)	0.051
LDH, max. [U/L], (IQR)	614.0 (413.0–1115.5)	818.0 (414.5–1334.3)	0.604	1.21 (0.81–1.81)	0.345	1.30 (0.87–1.94)	0.209
BNP [pg/mL], (IQR)	315.1 (176.5–1354.0)	1622.0 (80.6–6453.0)	0.656	1.31 (0.60–2.88)	0.499	1.23 (0.57–2.67)	0.605
CRP [mg/dL], (IQR)	0.6 (0.4–1.5)	1.7 (0.8–11.4)	**0.001**	1.92 (1.26–2.91)	**0.002**	1.83 (1.21–2.78)	**0.004**
Creatinine, pre-PCI [mg/dL], (IQR)	1.08 (0.98–1.33)	1.28 (0.94–1.76)	0.201	1.57 (1.03–2.41)	**0.037**	1.45 (0.94–2.25)	0.093
Creatinine, post-PCI [mg/dL], (IQR)	0.93 (0.79–1.06)	1.39 (0.78–1.76)	**0.026**	2.38 (1.54–3.68)	**<0.001**	2.19 (1.38–3.46)	**<0.001**
HbA1c [%], (IQR)	5.6 (5.2–6.2)	5.5 (5.3–6.2)	0.966	0.78 (0.28–2.31)	0.654	0.62 (0–2.29)	0.473

Metric data are displayed as median with the respective interquartile range (IQR) in parenthesis. Discrete data are demonstrated as counts with percentages in parenthesis. For statistical analysis, Mann–Whitney U test was used for metric data and Chi-square test was used for categorial data. Cox proportional hazard regression analysis was performed, the multivariate model was adjusted for age and sex. Statistical significance is shown in bold values. AF = atrial fibrillation, AMI = acute myocardial infarction, BMI = body mass index, BNP = B-type natriuretic peptide, BP = blood pressure, CI = confidence interval, CK = creatinine kinase, CKD = chronic kidney disease, CK-MB = creatinine kinase—muscle and brain type, CPR = cardio-pulmonary resuscitation, CRP = C-reactive protein, CS = cardiogenic shock, CVD = coronary vessel disease, CX = left circumflex artery, HbA1c = haemoglobin A1c, HR = hazard ratio, LAD = left anterior descending artery, LDH = lactate dehydrogenase, LM = left main, LVF = left ventricular function, PCI = percutaneous coronary intervention, Marg. = marginal branch, RCA = right coronary artery, STEMI = ST-elevation myocardial infarction, TnT = troponin T, VT = ventricular tachycardia, VF = ventricular fibrillation.

## Data Availability

Not applicable.

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
