# Peer review of "Cardiogenic Shock Does Not Portend Poor Long-Term Survival in Patients Undergoing Primary Percutaneous Coronary Intervention"

_jpm, 2022, doi:10.3390/jpm12081193_

Round 1
Reviewer 1 Report
The article under review represents a study investigating the impact of cardiogenic shock (CS) on long-term survival in patients after acute coronary syndrome. The authors conclude that CS was a strong prognostic factor for short-term survival but not long-term survival. Strengths of the study include a relatively large sample size and a well-written article. However, the following aspects should be considered:
· In the methods section, the authors write: “Immediately before and after coronary angiography, blood samples were collected and analyzed by local standards of the Department of Laboratory Medicine at the Medical University of Vienna (Austria).” Please describe briefly what types of lab tests were performed.
· Cardiovascular mortality was defined as the primary study endpoint. Were there any secondary endpoints?
· A Cox regression model was used to assess the influence of CS and prognostic factors on cardiovascular mortality. Has the proportional hazards assumption been checked?
· The authors mention that in the multivariate analyses, only adjustments for age and sex were made. Why were other potential confounders not considered?
· In the discussion, it is stated: “Especially in the era of personalized medicine, precise patient characterization and the detection of risk factors is of utmost importance to take a step towards individualized treatment strategies.” It would be beneficial to mention (as a future perspective) other recently published cardiovascular studies that have shown machine learning models as an effective way of detecting risk factors for ACS-related adverse events in a personalized way (as discussed in, for example, PMID: 34423350).
Reviewer 2 Report
The authors should be congratulated for the efforts to present the result of a retrospective study about long-term outcomes after PCI in patients with Cardiogenic shock. Although such studies have inherent limitations due to study design, they have useful information that long-term outcomes are comparable in CS and non-CS patients. And age, CKD, and, moderate to severely reduced LV function was highly predictive values to identify CS patients at risk for cardiovascular mortality during long-term follow-up.
